# Assessment of Osteoporosis and Vitamin D3 Deficiency in Patients with Idiopathic Benign Paroxysmal Positional Vertigo (BPPV)

**DOI:** 10.3390/medicina59050862

**Published:** 2023-04-28

**Authors:** Katarzyna Miśkiewicz-Orczyk, Wojciech Pluskiewicz, Beata Kos-Kudła, Maciej Misiołek

**Affiliations:** 1Department of Otorhinolaryngology and Laryngological Oncology, Faculty of Medical Sciences in Zabrze, Medical University of Silesia in Katowice, 41-800 Zabrze, Poland; 2Department and Clinic of Internal Diseases, Diabetology, and Nephrology, Metabolic Bone Diseases Unit, Faculty of Medical Sciences in Zabrze, Medical University of Silesia in Katowice, 41-800 Zabrze, Poland; 3Department of Endocrinology and Neuroendocrine Tumors, Department of Pathophysiology and Endocrinology, Medical University of Silesia, 40-055 Katowice, Poland

**Keywords:** benign paroxysmal positional vertigo, bone mineral density, 25(OH) vitamin D

## Abstract

*Background and objectives*: Osteoporosis and vitamin D3 deficiency may be risk factors of benign paroxysmal positional vertigo (BPPV). The aim of this study was to assess the prevalence of osteoporosis and 25(OH) vitamin D3 deficiency in a group of patients with idiopathic benign paroxysmal positional vertigo. *Materials and Methods*: Thirty-five patients (twenty-eight women and seven men) with posterior semicircular canal BPPV were enrolled in the study. The subjects underwent hearing assessment (tonal audiometry and impedance audiometry) and the Dix-Hallpike maneuver. Serum 25(OH) vitamin D3 levels were determined and lumbar spine bone densitometry was performed. The relationships between sex, age, height, Body Mass Index (BMI), vitamin D3 levels and bone densitometry results were assessed. *Results*: The diagnosis of osteoporosis was confirmed in 1 patient (3%), 3 subjects were osteopenic (8.6%), and normal bone densitometry was found in 31 (88.6%) patients. *Conclusions*: We found no statistically significant relationships between age, BMI or vitamin D3 levels and bone densitometry results in patients with idiopathic BPPV.

## 1. Introduction

Benign paroxysmal positional vertigo (BPPV) is the most common cause of peripheral vertigo among people worldwide [1,2,3]. Canalolithiasis and cupulolithiasis underlie the pathophysiology of the disease [4,5]. Canalolithiasis comprises floating otoconial debris within the endolymph of a semicircular canal. In turn, cupulolithiasis comprises free-floating otoconial debris adherent to the cupula.

In both cases, otoliths stimulate the cells of the cupula of the semicircular canal. Positional vertigo is generated in the posterior semicircular canal and the lateral semicircular canal (canalolithiasis), as well as in the anterior semicircular canal (cupulolithiasis). In each case, pathological stimulation of the cupula by endolymph movement or otoliths generates short paroxysmal vertigo that occurs with changes in the body position and is not associated with hearing loss or tinnitus [6]. 

According to the guidelines of The Japan Society for Equilibrium Research (2017), benign paroxysmal positional vertigo is diagnosed when the patient presents with positive maneuvers, i.e., the presence of nystagmus typical of a particular semicircular canal with subjective vertigo [7]. The most common factors that increase the risk of BPPV include sex (female gender), hypertension, diabetes mellitus, migraine or head trauma in the past [8,9,10,11]. Less common factors that increase the risk of recurrent BPPV include vitamin D3 deficiency and osteoporosis, which were confirmed by Chen et al. in group of 3060 patients [12]. Jeong et al. confirmed that vitamin D and calcium supplementation reduce recurrences of BPPV [13]. He at al. provided stronger research that patients with BPPV were associated with a lower T-score and a higher risk of osteoporosis and osteopenia. Their results demonstrated that lower bone mineral density may be a risk factor for BPPV [14]. Jeong et al. indicated that only the existence of osteopenia/osteoporosis was associated with an increased risk of BPPV [15]. Their study confirmed that in a group of women aged > or =45 years, the lowest T-scores were also decreased in the recurrent BPPV group, compared with those in the de novo group. Despite this, the results of many authors are still controversial. 

Osteoporosis is a disease characterized by low bone mineral density and disorders of skeletal microarchitecture [16,17]. Bone fragility is affected by remodeling and leads to bone brittleness and susceptibility to pathological fractures of the wrist or hip. Processes occurring in a bone structure may affect the function of vestibular organs and increase the risk of vertigo, falls and pathological fractures [18,19,20]. Osteoporosis and vitamin D deficiency may derange calcium metabolism, which affects the otolith structure, and may increase the risk of it detaching from the utricle area and floating to semicircular canals [21,22]. 

The aim of this study was to assess the level of vitamin D3 expressed as 25(OH) and the prevalence of osteoporosis in patients with idiopathic BPPV, without any risk factors based on the relationships between age, BMI, vitamin D3 levels and bone densitometry measurements.

## 2. Materials and Methods

### 2.1. Population and Sample Collection

Thirty-five patients with idiopathic BPPV were enrolled in the study.

Audiological assessment (i.e., tonal audiometry and impedance audiometry) and the Dix-Hallpike maneuver were performed to confirm horizontal semicircular canal BPPV in the Department of Otorhinolaryngology and Oncological Laryngology, Medical University of Silesia, Katowice, Poland. 

Serum 25(OH) vitamin D3 levels were measured, and bone densitometry results of the lumbar spine were performed in the Department and Clinic of Internal Diseases, Diabetology, and Nephrology, Metabolic Bone Diseases Unit, Faculty of Medical Sciences in Zabrze, Medical University of Silesia in Katowice, Zabrze. The study was conducted from October to December 2022.

### 2.2. Inclusion Criteria

The study included patients between 18 and 65 years of age with idiopathic BPPV confirmed by a positive Dix-Hallpike test and normal hearing (normal tonal audiometry) and normal tympanic pressure confirmed by impedance audiometry (i.e., type A tympanogram). Patients without risk factors for BPPV, such as a history of head trauma, migraine, osteoporosis, vitamin D3 deficiency or other conditions of the peripheral part of the vestibular organ (e.g., Meniere’s disease and neuronitis vestibularis), were included in the study. Patients had no past history of vertigo and when it occurred it was not recurrent. 

### 2.3. Exclusion Criteria

The exclusion criteria were as follows: age below 18 years and above 65 years, causes of vertigo other than idiopathic BPPV (e.g., migraine, head trauma, Meniere’s disease, neuronitis vestibularis or central vertigo), conductive and/or sensorineural unilateral or bilateral hearing loss, abnormal impedance audiometry (except for the type A tympanogram) and other causes that could affect the bone status and/or could be risk factors for BPPV (e.g., hypertension; endocrine disease; bowel, kidney or liver diseases; rheumatoid arthritis; and chronic use of some medications, including steroids, anticonvulsants, proton pump inhibitors, anticoagulants or loop diuretics)**.**

### 2.4. Audiological Tests

In the audiometric test room, tonal audiometry and impedance audiometry were performed. Tonal audiometry was performed using an AD229e audiometer with the use of the ascending method for air conduction (frequency range 250–8000 Hz) and bone conduction (frequency range 250–4000 Hz) separately for the right and the left ear. The compliance and pressure in the external auditory canal and eardrum cavity were measured using an AT235 tympanometer. The pressure ranging from −100 daPa to +100 daPa was considered normal, while the range of 0.3–1.3 mL was regarded as the normal compliance of the ear conduction system.

### 2.5. Assessment of Vitamin D3 Levels

Electrochemiluminescence (ECLIA) was used to determine 25-OH vitamin D3 concentrations in the analytical laboratory after collecting blood serum (3 cm^3^). The range of 30–80 ng/mL was considered normal. 

### 2.6. The Dix-Hallpike Maneuver

The Dix-Hallpike maneuver was performed by rapid moving of the patient from the sitting to the supine position with the head down and tilted to the right (45°). After the return to the sitting position, the maneuver was performed in the same sequence with the head tilted to the left. The presence of positioning nystagmus and subjective vertigo reported by the patient were considered a positive result. 

### 2.7. Bone Densitometry in the Lumbar Spine

Lumbar spine bone densitometry results were performed in the densitometry laboratory (Hologic Explorer densitometer; S/N 91755). The patient was in the supine position with the lower extremities raised and flexed. The dose of ionizing radiation in each patient was below 1 microSv. The examination was conducted to evaluate the lumbar spine at L1–L4. Bone mineral density (BMD) and the Z-Score for L1–L4 were measured. A Z-Score ≥ −2 was considered normal. In postmenopausal women, T-score was used for interpretation of bone densitometry measurements.

### 2.8. Statistical Analysis

The statistical analysis was conducted using the IBM SPSS Statistics 25 to verify the research hypotheses. We performed basic descriptive statistics, Shapiro–Wilk tests, Mann–Whitney U tests, Pearson’s r correlation analysis and Spearman’s rank correlation analysis. Mann–Whitney U tests were conducted in the analysis of correlation between sex and bone densitometry due to the significant difference in the sizes of the groups. The *p* value in Table 1 and Table 2 shows the result of Shapiro–Wilk test. If the result is non-significant, the variable distribution is similar to normal distribution. If it is significant, the skewness is taken into consideration. Because all skewness values are within the range as suggested by George and Mallery, all analyses were performed by means of parametrical statistics, and thus we presented mean and standard deviation and not quartiles. 

A *p*-value < 0.05 was considered statistically significant.

## 3. Results

### 3.1. Patient Clinical Characteristics 

From October to December 2022, 35 patients (28 women, 80%; 7 men, 20%) aged between 23 and 65 years (M = 46.57; SD = 11.32) participated in the study. Eight postmenopausal women were included, and all men were younger than 50 years. The weights of the patients ranged from 47 to 111.6 kg (M = 73.99; SD = 18.48). The heights reported by the patients ranged from 152 to 192 cm (M = 166.83; SD = 9.90), while the heights assessed by the technician ranged from 151.4 to 191.7 cm (M = 165.76; SD = 9.74). The BMIs of the patients ranged from 17.81 to 39.22 (M = 26.66; SD = 4.85).

A 25(OH) Vitamin D3 deficiency was found in 19 (54%) women and 2 men (6%). Osteoporosis was confirmed in one postmenopausal patient aged 63 (3%) (Z-Score: −1; T-score: −2.7). In three postmenopausal women, the T-score was in the osteopenic range (8.6% of whole group). The remaining patients (N = 31; 88.6%) had normal bone densitometry results. 

Table 1 shows the descriptive results of the variables, i.e., age, height, Body Mass Index (BMI), 25(OH) vitamin D3 levels and bone densitometry results in the women. A total of 28 women aged between 29 and 65 years (M = 49; SD = 10.45) participated in the study. The weights of the women ranged from 47 to 102.3 kg (M = 68.08; SD = 14.11). The heights reported by the patients ranged from 152 to 178 cm (M = 163.43; SD = 7.23), while the heights assessed by the technician ranged from 151.4 to 174.4 cm (M = 162.54; SD = 7.18). The BMIs of the women ranged from 17.81 to 39.22 (M = 25.70; SD = 4.66). The 25(OH) vitamin D3 levels ranged from 14.48 to 66.86 ng/mL (M = 32.63; SD = 12.70). The total BMD in the women ranged from 0.75 to 1.21 (M = 0.96; SD = 0.12). The total Z-score ranged from −1.60 to 2.40 (M = 0.05; SD = 1.08). The total T-score ranged from −13 to 1.5 (M = −1.16; SD = −0.95).

Table 2 shows the descriptive results of the variables, i.e., age, height, Body Mass Index (BMI), 25(OH) vitamin D3 levels and bone densitometry results in the men. Seven men aged between 23 and 45 years (M = 36.86; SD = 9.87) participated in the study. The weights of the men ranged from 74 to 114.6 kg (M = 97.63; SD = 14.97). The heights reported by the patients ranged from 172 to 192 cm (M = 180.43; SD = 7.11), while the heights assessed by the technician ranged from 169.5 to 191.7 cm (M = 178.66; SD = 7.97). The BMIs of the men ranged from 25.76 to 36.79 (M = 30.50; SD = 3.71). The 25(OH) vitamin D3 levels ranged from 17.73 to 34.31 ng/mL (M = 25.03; SD = 5.09). The total BMD in the men ranged from 0.96 to 1.16 (M = 1.06; SD = 0.08). The total Z-score ranged from −1.20 to 0.80 (M = −0.13; SD = 0.69). The total T-score ranged from −1.20 to 0.70 (M = −0.26; SD = 0.71).

### 3.2. The Relationships between Age, Height, BMI, 25(OH) Vitamin D3 Levels and Bone Densitometry Results

The relationships between age, height, BMI, 25(OH) vitamin D3 levels and bone densitometry results were analyzed. Pearson’s correlations were performed. We found one statistically significant correlation. Height measured by the technician correlated positively with total BMD level (Figure 1). The total BMD level rose with the height of the patients. The observed correlation was moderately strong. We also found two correlations on the level of statistical tendency. The total BMD level was correlated negatively with the age of the patients, while the total Z-score level was correlated positively. Older patients tended to have a lower BMD level and higher Z-score. Both correlations were weak. We also observed no correlation between total T-score and age, height, BMI or 25(OH) vitamin D3 level. Other correlations shown in Table 3 were not statistically significant. 

## 4. Discussion

Benign paroxysmal positional vertigo (BPPV) is the most common cause of peripheral vertigo that affects almost 17–42% of all patients with vertigo [23,24]. The disease is characterized by a short-term spinning sensation upon changes in the body position [1,25]. Attacks of vertigo are not accompanied by any auditory sensation. This disease is observed in the posterior semicircular canal in almost 95% of cases [2,3]. Not only head trauma, but also viral infection or ototoxic drugs may have an influence on otoliths’ migration. In addition, middle ear surgery or an autoimmunity are suspected to be involved in this process [26,27]. Sudden attacks of severe vertigo without coexistence of auditory symptoms, such as hearing loss or tinnitus, can evoke in patients not only anxiety and fear but also even obsessive-compulsive disorder [28]. These can be reasons why patients begin to avoid critical body positions. Patients with BPPV have an increased risk of falls and injury. The increased risk of falls is exacerbated by the fear of moving and leaving the house, which results in the isolation of patients from society [29]. Therefore, it is crucial to assess the risk factors for BPPV to prevent the condition and to raise awareness among patients and doctors about the occurrence of the disease. 

The relationship between BPPV and osteoporosis is still controversial and difficult to prove. On the one hand, the authors qualified to their research heterogeneous group of patients who take different drugs (e.g., steroids, anticonvulsants, proton pump inhibitors, anticoagulants or diuretics) and who suffer from civilization diseases such as hypertension, endocrine disease, bowel, kidney and liver diseases, or rheumatoid arthritis. On the other hand, the authors included in their research patients with conductive and/or sensorineural unilateral or bilateral hearing loss, abnormal impedance audiometry, and patients with migraines, Ménière’s disease or neuronitis vestibularis or who have experienced head trauma. We have to remember that all of those factors can affect the bone status and/or could be risk factors for BPPV. For example, De Stefano et al. confirmed that the combination of two or more diseases (e.g., hypertension, diabetes, osteoarthrosis, osteoporosis and depression) further increases the risk of relapsing BPPV, worsened by the presence of osteoporosis [30]. In addition, Zhu et al. reported that Ménière’s disease, hypertension, migraine and hyperlipemia may be independent risk factors for the recurrence of BPPV [31]. 

The aim of this paper was to evaluate 25(OH) vitamin D levels and bone densitometry results in a group of patients with idiopathic BPPV. According to some authors, risk factors for idiopathic BPPV may include degenerative changes that occur in the otoliths in the utricle [22]. In turn, Fu et al. reported that poor physical activities and prolonged recumbent position time could be important risk factors for BPPV patients (OR = 18.92, 95% CI: 6.34–56.43, *p* = 0.00 and OR = 1.15, 95% CI: 1.01–1.33, *p* < 0.04) [32]. Young patients (aged 18–65 years) enrolled in this study were professionally active and did not have a past history of vertigo. They did not report BPPV risk factors, but most of them (60%) had vitamin D3 deficiency. However, in this group vitamin D3 levels alone had no statistically significant effect on bone densitometry results. This is in line with the reports of other authors who showed a vitamin D3 deficiency in patients with not idiopathic but recurrent BPPV [33,34,35]. We have to indicate that most of this research concerns the issues of recurrent BPPV. This is opposite to our results of idiopathic BPPV, which occurred for the first time in our patients. For example, on a large meta-analysis on a group of 1254 patients, Chen et al. confirmed the relationship between low vitamin D3 levels and recurrent BPPV compared to a control group (MD = −2.12; 95% CI, −3.85 to −0.38; *p* = 0.02) [12]. In addition, Seong-Hae et al. confirmed a statistically significant effect of vitamin D supplementation in patients with recurrent BPPV compared to the controls (0.83 (95% CI, 0.74–0.92) vs. 1.10 (95% CI, 1.00–1.19)) recurrences per one person-year [13]. This shows how important it is to measure the level of vitamin D among patients with vertigo, especially in the context of the prevention of vertigo in young and professionally active patients. On the other hand, Talaat et al. published interesting results on a group of 80 patients with idiopathic BPPV [36]. The authors indicated that low levels of vitamin D were related to the development of BPPV and very low levels were associated with recurrence of this disease. Wu et al. reported that a decreased 25(OH) vitamin D level in early-onset female patients may increase the odds of attacks of vertigo 1 week after successful repositioning maneuvers [37]. This study shows how important the role of vitamin D is in the treatment of BPPV, even after successful performance of the release maneuvers. Such findings were also confirmed by other authors [38,39]. In turn, vitamin D3 deficiency may decrease estrogen levels, which increases the risk of BPPV in women over 65 years of age [11,40]. This group of patients is particularly at risk of falls and, consequently, pathological fractures. In a study on mice, Yang et al. reported the degeneration of otoliths in the saccule and utricle when estradiol deficiency was reported [41]. On the other hand, Han et al. reported that there were no significant differences in bone metabolism in postmenopausal female patients with different types of idiopathic BPPV [42]. Our group of patients included young subjects (M = 46.57; SD = 11.32) and only eight women were postmenopausal, which may explain the results obtained in our study. No relationships were found between sex, age, BMI, 25(OH) vitamin D3 levels and bone densitometry results. This could be caused by the fact that most patients had normal bone densitometry results (60%) and osteopenia (37%). This confirms the reports of other authors who did not find osteopenia to be a statistically significant risk factor for BPPV [12,43]. 

At the same time, many authors have stressed the role of osteoporosis as a recurrent factor in BPPV [43,44]. Once again, we have to indicate that these authors focused on patients with recurrent BPPV, not like our study of first-time BPPV, without recurrence in the past. This could be one reason why our research did not confirm that result. In a large meta-analysis on 22750 patients, Li et al. found osteoporosis to be a risk factor for recurrent BPPV (OR 1.385; 95% CI, 1.287–1.491) [8]. Similar findings were confirmed by Chen et al. (OR = 1.72; 95% CI, 1.03–2.88; *p* = 0.04) [12]. On the other hand, Kim et al., in a group of 198 patients with idiopathic BPPV, noticed that bone mineral density was markedly decreased but there were no significant differences according to BPPV recurrence [45]. Yang et al., in a group of women with idiopathic BPPV, showed a significantly decreased BMD but no association with recurrence of BPPV in this group of patients [46]. This is in line with our research results. On the other hand, Chan et al. reported that a group of 6649 patients with osteoporosis were found to have a 1.82-fold higher risk of developing BPPV than those without this disease [47]. We have to remember how diverse that group of patients was. They suffered from many civilization diseases, for example hypertension or diabetes mellitus, which could affect the bone status. Osteoporosis increases the risk of falls and fractures of the femoral neck or wrists [48,49]. Together with vertigo, the risk of injury, the fear of injury that causes isolation of patients from society and the costs associated with the treatment of injuries increase significantly [50,51]. Therefore, it is crucial to be aware of the risk factors for diseases which manifest as vertigo, and which may be associated with many factors that burden the health care system. It is all the more important, since they affect young and professionally active individuals.

## 5. Conclusions

The statistical analysis in this paper did not show relationships between age, BMI, 25(OH) vitamin D3 levels and densitometry results in the study group of 35 patients with idiopathic BPPV. Further studies on larger cohorts are warranted to confirm our findings. 

## Figures and Tables

**Figure 1 medicina-59-00862-f001:**
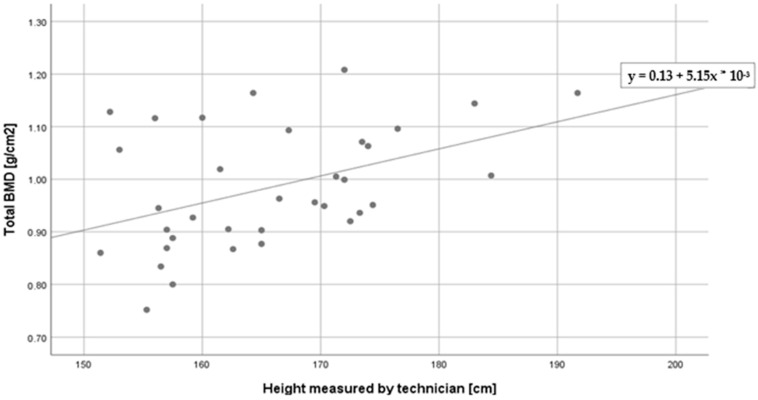
Correlation between total BMD (g/cm^2^) and height measured by technician (cm).

**Table 1 medicina-59-00862-t001:** Age, height, BMI, 25(OH) vitamin D3 levels and bone densitometry results in women (N = 28).

	M	Me	SD	Sk.	Kurt.	Min.	Max.	W	*p*
Age (years)	49	50	10.45	−0.35	−1.04	29	65	0.94	0.134
Weight (kg)	68.08	63.65	14.11	0.73	−0.09	47	102.3	0.94	0.122
Height reported by the patient (cm)	163.43	162	7.23	0.36	−0.85	152	178	0.96	0.325
Height measured by the technician (cm)	162.54	161.85	7.18	0.27	−1.19	151.4	174.4	0.93	0.079
BMI (kg/m^2^)	25.70	25.32	4.66	0.78	1.14	17.81	39.22	0.96	0.292
25(OH) Vit. D (ng/mL)	32.63	29.21	12.70	1.34	1.64	14.48	66.86	0.86	0.001
Total BMD (g/cm^2^)	0.96	0.94	0.12	0.42	−0.55	0.75	1.21	0.96	0.345
Total Z-score	0.05	−0.05	1.08	0.36	−0.74	−1.60	2.40	0.96	0.397
Total T-score	−1.16	−0.95	2.54	−3.91	18.59	−13	1.5	0.58	<0.001

M—mean; Me—median; SD—standard deviation; Sk.—skewness; Kurt.—kurtosis; Min and Max—the lowest and highest values of the distribution; W—result of the Shapiro–Wilk test; *p*—significance.

**Table 2 medicina-59-00862-t002:** Age, height, BMI, 25(OH) vitamin D3 levels and bone densitometry results in men (N = 7).

	M	Me	SD	Sk.	Kurt.	Min.	Max.	W	*p*
Age (years)	36.86	41	9.87	−0.92	−1.20	23	45	0.78	0.026
Weight (kg)	97.63	101	14.97	−0.44	−0.90	74	1140.6	0.95	0.696
Height reported by the patient (cm)	180.43	178	7.11	0.59	−0.78	172	192	0.94	0.666
Height measured by the technician (cm)	178.66	176.50	7.97	0.59	−0.77	169.5	1910.7	0.94	0.636
BMI (kg/m^2^)	30.50	30.36	3.71	0.60	0.07	25.76	360.79	0.97	0.929
25(OH) Vit. D (ng/mL)	25.03	23.87	5.09	0.71	1.79	17.73	340.31	0.94	0.683
Total BMD (g/cm^2^)	1.06	1.07	0.08	0	−1.56	0.96	10.16	0.95	0.697
Total Z-score	-0.13	0	0.69	−0.23	−0.61	−1.20	00.80	0.98	0.944
Total T-score	-0.26	−0.20	0.71	0.79	1.59	−1.20	00.70	0.94	0.650

M—mean; Me—median; SD—standard deviation; Sk.—skewness; Kurt.—kurtosis; Min and Max—the lowest and highest values of the distribution; W—result of the Shapiro–Wilk test; *p*—significance.

**Table 3 medicina-59-00862-t003:** The relationships between age, height, BMI, 25(OH) vitamin D levels and bone densitometry results (N = 35).

	Age (Years)	Height Measured by the Technician (cm)	BMI (kg/m^2^)	Vit. D (ng/mL)
Total BMD (g/cm^2^)	−0.30 ^	0.44 **	0.20	0.03
Total Z-score	0.29 ^	0.15	0.25	0.25
Total T-score	0.27	0.13	0.18	0.12

**—*p* < 0.01; ^—0.05 < *p* < 0.1.

## Data Availability

All data involved in this study will be made available by the corresponding author upon request.

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
