# Peer review of "Assessment of Osteoporosis and Vitamin D3 Deficiency in Patients with Idiopathic Benign Paroxysmal Positional Vertigo (BPPV)"

_medicina, 2023, doi:10.3390/medicina59050862_

Round 1
Reviewer 1 Report
The authors present an interesting paper on the prevalence of osteoporosis and Vitamin D3 deficiency in patients with idiopathic benign paroxysmal positional vertigo. However, the study is limited by the small sample size and lack of controls which impedes the authors from finding any associations.
Several comments are listed below.
· The abstract lacks an introduction.
· In the Introduction section, the authors state that osteoporosis and Vit D deficiency might increase the risk of BPPV. However, they do not state why/how this would result. This needs to be discussed in much more detail. In the discussion section, they mention the link between osteoporosis and propensity for falling which in turn increases the risk of BPPV. The authors need to be clearer on the nature of the relationship of osteoporosis with BPPV.
· Over how many months did recruitment take place? 35 is unfortunately a very small number to derive any significant observations. Furthermore, no controls were included in this study. Was there a reason why age-matched controls were not recruited?
· Why was only the lumbar spine BMD measured? Is the hip not routinely measured? Especially since the LS is more prone to errors (e.g., osteophytes, etc). Perhaps since the individuals were younger, the authors could also have considered measuring the wrist. A full body DXA would have been the most ideal.
· The authors keep referring to bone densitometry as their phenotype when it should be bone mineral density. This should be even in their keywords (instead of bone densitometry). Same for Vitamin D3 (ideally 25-OH Vitamin D).
· Why did the authors only include newly diagnosed BPPV cases? What was the reason behind the exclusion of known BPPV cases?
· The authors only used the Z-score. However, the T-score should be used in men aged 50 years and above, and post-menopausal women. It would be best to quote both in the study.
· A Z-score of ≥ -2.0 is not normal. There is also an osteopenic range when using the Z-score. In fact, authors do not mention anyone having osteopenia in the manuscript expect towards the end of the Discussion section which is confusing. Osteopenia should not be overlooked and others studies have taken this into consideration.
· The results section is poorly presented. How old were the recruited women? How old were the men? A table showing the population characteristics in men and women separately and in combination should be shown, including the age, BMD, T-score, Z-score, BMI, 25-OH Vit D, etc.
Did the authors collect any more information from the participants? Did they perform an interviewer-led questionnaire on lifestyle factors, etc?
· If the data is non-normally distributed, there is no need to show the mean and standard deviation. Instead, the median with the 25th and 75th percentiles is enough (Table 1).
· The authors state that osteoporosis was present in one postmenopausal woman (Z-score = -1.0). This would mean that her BMD is normal not osteoporotic if the Z-score was -1.0. In the same paragraph, the authors state that the rest of the recruited individuals had a normal BMD when this doesn’t seem to be the case.
How many of the recruited women were postmenopausal?
· Section 3.2 is confusing. What are the authors trying to show in Table 2? What are the associations? It would have made more sense to graphically display the data e.g., using box plots or scatterplots. What does ^ represent?
· Did the authors consider seasonal variability in the case of Vitamin D?
· What about the levels of PTH and calcium in these recruited individuals? What about kidney function?
· In the Discussion section, the authors also state that the participants were ‘professionally active’. What does this mean?
· The biggest limitation of the study is the sample size and the authors should stress this even more as their limitation.
Author Response
Dear Editor!
We would like to thank You for valuable comments.
Belowe, the changes that we made to the manuscript (data marked in red in manuscript).
Best Regards!
Katarzyna Miśkiewicz-Orczyk & team.
- The abstract lacks an introduction.
Answer: We added information about osteoporosis and BPPV, we are limited by amount of words in the introduction. .
- In the Introduction section, the authors state that osteoporosis and Vit D deficiency might increase the risk of BPPV. However, they do not state why/how this would result. This needs to be discussed in much more detail. In the discussion section, they mention the link between osteoporosis and propensity for falling which in turn increases the risk of BPPV. The authors need to be clearer on the nature of the relationship of osteoporosis with BPPV.
Answer: We added some information about relationship between BPPV and osteoporosis.
- Over how many months did recruitment take place? 35 is unfortunately a very small number to derive any significant observations. Furthermore, no controls were included in this study. Was there a reason why age-matched controls were not recruited?
Answer: We recruited patients from October to December 2022. We enrolled 96 patients with BPPV confirmed by a positive Dix-Hallpike test. Most of their had sensorineural hearing loss or hypertension, used many drugs. The aim of our study was to investigate vitamin D level and bone densitometry only in patients diagnosed for the first time (no past history of vertigo, no hearing loss) and without risk factors such as a history of head trauma, migraine, osteoporosis, vitamin D3 deficiency or other conditions of the peripheral part of the vestibular organ (e.g. Meniere’s disease, neuronitis vestibularis); other causes that could affect the bone status and/or could be risk factors for BPPV (e.g. hypertension, endocrine disease, bowel, kidney, liver diseases, rheumatoid arthritis, chronic use of some medications, including steroids, anticonvulsants, proton pump inhibitors, anticoagulants, or loop diuretics). We mentioned about this in exclusion criteria. That is why our group counts only 35 patients.
- Why was only the lumbar spine BMD measured? Is the hip not routinely measured? Especially since the LS is more prone to errors (e.g., osteophytes, etc). Perhaps since the individuals were younger, the authors could also have considered measuring the wrist. A full body DXA would have been the most ideal.
Answer: Generally, in order to establish bone status in adults hip and lumbar spine are measured. However, in younger population as our patients spine BMD is more appropriate because bone mass of the hip usually in mid age individuals is within normal range. Of course forearm BMD might be measured but this skeletal site is rather chosen as a place for skeletal assessment when spine and hip cannot be established (spine degeneration, hip surgery at both sites, high body mass). The lack of hip measurement was added this point as a limitation of the study.
- The authors keep referring to bone densitometry as their phenotype when it should be bone mineral density. This should be even in their keywords (instead of bone densitometry). Same for Vitamin D3 (ideally 25-OH Vitamin D).
Answer: We agree with these remarks and bone densitometry was replaced by bone mineral density. The same changes were done for vitamin D. In order to clarify vitamin D3 status we mentioned in revised manuscript (part Introduction) that the word vitamin D3 in the whole text was expressed by measurement of 25(OH)D3. We added bone mineral density and 25(OH) Vitamin D to our keywords. We added 25(OH) D3 Vit. and “densitometry results” in text.
- Why did the authors only include newly diagnosed BPPV cases? What was the reason behind the exclusion of known BPPV cases?
Answer: We were curious if that group of patients (not burdened with additional diseases, hearing loss and previous attacks of vertigo) had that risk factors (vit. D deficiency or osteoporosis).
- The authors only used the Z-score. However, the T-score should be used in men aged 50 years and above, and post-menopausal women. It would be best to quote both in the study.
Answer: In current study we had no men older than 50 years and 8 postmenopausal women. We added that information to patients characteristics.
- A Z-score of ≥ -2.0 is not normal. There is also an osteopenic range when using the Z-score. In fact, authors do not mention anyone having osteopenia in the manuscript expect towards the end of the Discussion section which is confusing. Osteopenia should not be overlooked and others studies have taken this into consideration.
Answer: The use of bone densitometry is different in postmenopausal women and men older that 50 year and in premenopausal women and younger men. Only in postmenopausal women and men older that 50 year we may diagnose osteoporosis when T-score for either spine or hip is below -2.5. In other patients in order to describe bone status in other subjects we must use only Z-score, and this value is below -2.0 one may consider that bone status is abnormal. In such subjects a term ‘osteoporosis’ or ‘osteopenia’ cannot be used. In current study we had no men older than 50 years and 8 postmenopausal women. Only one women aged 63 years had T-score below -2.5. We corrected a mistake in previous version of manuscript in regard to this patient. The clear description of bone densitometry is added to manuscript text.
- The results section is poorly presented. How old were the recruited women? How old were the men? A table showing the population characteristics in men and women separately and in combination should be shown, including the age, BMD, T-score, Z-score, BMI, 25-OH Vit D, etc.
Answer: In current study we had no men older than 50 years and 8 postmenopausal women. We divided group into two, we presented results for both group separately (Table 1, Table 2).
. Did the authors collect any more information from the participants? Did they perform an interviewer-led questionnaire on lifestyle factors, etc?
Answer: In all patients we performed interview and physical examination.
- If the data is non-normally distributed, there is no need to show the mean and standard deviation. Instead, the median with the 25thand 75th percentiles is enough (Table 1).
Answer: Because all skewness values are within range as suggested by George and Mallery, all analysis were perfomed by means of parametrical statistics, thus we presented mean and standard deviation and not quartiles.
- The authors state that osteoporosis was present in one postmenopausal woman (Z-score = -1.0). This would mean that her BMD is normal not osteoporotic if the Z-score was -1.0. In the same paragraph, the authors state that the rest of the recruited individuals had a normal BMD when this doesn’t seem to be the case.
Answer: We decided to present the results of whole group using Z-score because the majority of subjects were premenopausal women and men younger than 50 years. In other postmenopausal women only 3 had T-score within osteopenic range and 4 had normal T-score value. Therefore, in order to obtain clear view of the results and their interpretation we decided to use Z-score. Please also note that only 1 patient was osteoporotic.
How many of the recruited women were postmenopausal?
Answer: In current study we had 8 postmenopausal women.
- Section 3.2 is confusing. What are the authors trying to show in Table 2? What are the associations? It would have made more sense to graphically display the data e.g., using box plots or scatterplots. What does ^ represent?
Answer: We added information about associations and Figure 1.
- Did the authors consider seasonal variability in the case of Vitamin D?
Answer: We did not consider this marker.
- What about the levels of PTH and calcium in these recruited individuals? What about kidney function?
Answer: We did not measured PTH and calcium, all patients had normal kidney function (interview data).
- In the Discussion section, the authors also state that the participants were ‘professionally active’. What does this mean?
Answer: It means that patients were not retired.
- The biggest limitation of the study is the sample size and the authors should stress this even more as their limitation.
Answer: We added that information to the text.
Reviewer 2 Report
The paper is interesting, but requires to be improved:
- in the introduction the link between benign paroxysmal positional vertigo and osteoporosis should be clarified
- the control group should be better described, including the number of subjects involved
- it is very difficult include all patients in one group, it is known that BMD is different according to age, thus a better analysis can be performed maybe considering 2 groups. It will be interesting to have a detailed table with all values of BMD for each patient
- in table 1, it is not clear the p value reference. Values for controls should be provided.
Author Response
Dear Editor!
We would like to thank You for valuable comments.
Belowe, the changes that we made to the manuscript (data marked in red in manuscript).
Best Regards!
Katarzyna Miśkiewicz-Orczyk & team.
- in the introduction the link between benign paroxysmal positional vertigo and osteoporosis should be clarified
Answer: In the introduction we added information about relationship between BPPV and osteoporosis.
- the control group should be better described, including the number of subjects involved
Answer: In our study, we did not have a controll group. We wrote about that at the end of Discussion.
- it is very difficult include all patients in one group, it is known that BMD is different according to age, thus a better analysis can be performed maybe considering 2 groups. It will be interesting to have a detailed table with all values of BMD for each patient
Answer: We devided groups of patients into two (Table 1 and Table 2) and we presented them separately.
- in table 1, it is not clear the p value reference. Values for controls should be provided.
Answer: P value in the Table 1 shows the result of Shapir-Wilk test. If the result is non-significant, the variable distribution is similiar to normal dystribution. If it is significant, the skewness is taken into the consideration.
Round 2
Reviewer 1 Report
Authors present an improved version of the manuscript.
Author Response
Dear Editor!
We would like to thank You once again for Your work.
We added T-score value to Table 1,2 and 3.
We added more referencies to Discussion according to Editorial Office.
Best Regards
Katarzyna Miśkiewicz-Orczyk & Team

Reviewer 2 Report
The authors addressed some of my concerns, but it is important to underline that in adults for BMD evaluation T-score is used. Thus, if possible the value should be reported and the analysis updated.
Author Response
Dear Editor!
We would like to thank once again for Your work and valuable comments.
Belowe, the changes that we made to the manuscript (marked in red). We added T-score value to text and to Table 1,2 and 3.
We added more references to Discussion (according to Editorial Office).
Best Regards!
Katarzyna Miśkiewicz-Orczyk & team.
